# Crystal structure of an RNA/DNA strand exchange junction

Joshua C. Cofsky[1], Gavin J. Knott[2], Christine L. Gee[1,3,4], Jennifer A. Doudna[1,3,4,5,6,7,8]*

**1** Department of Molecular and Cell Biology, University of California, Berkeley, Berkeley, California, United States of America, **2** Department of Biochemistry & Molecular Biology, Monash Biomedicine Discovery Institute, Monash University, Clayton, VIC, Australia, **3** California Institute for Quantitative Biosciences (QB3), University of California, Berkeley, Berkeley, California, United States of America, **4** Howard Hughes Medical Institute, University of California, Berkeley, Berkeley, California, United States of America, **5** Department of Chemistry, University of California, Berkeley, Berkeley, California, United States of America, **6** Molecular Biophysics and Integrated Bioimaging Division, Lawrence Berkeley National Laboratory, Berkeley, Berkeley, California, United States of America, **7** Innovative Genomics Institute, University of California, Berkeley, Berkeley, California, United States of America, **8** Gladstone Institutes, University of California, San Francisco, San Francisco, California, United States of America

* doudna@berkeley.edu

**Data Availability Statement:** The data and model have been deposited in the PDB under accession code 7THB (https://doi.org/10.2210/pdb7THB/pdb). The diffraction images have been deposited in the IRRMC and can be found using the PDB accession code.

## Abstract

Short segments of RNA displace one strand of a DNA duplex during diverse processes including transcription and CRISPR-mediated immunity and genome editing. These strand exchange events involve the intersection of two geometrically distinct helix types—an RNA:DNA hybrid (A-form) and a DNA:DNA homoduplex (B-form). Although previous evidence suggests that these two helices can stack on each other, it is unknown what local geometric adjustments could enable A-on-B stacking. Here we report the X-ray crystal structure of an RNA-5′/DNA-3′ strand exchange junction at an anisotropic resolution of 1.6 to 2.2 Å. The structure reveals that the A-to-B helical transition involves a combination of helical axis misalignment, helical axis tilting and compression of the DNA strand within the RNA:DNA helix, where nucleotides exhibit a mixture of A- and B-form geometry. These structural principles explain previous observations of conformational stability in RNA/DNA exchange junctions, enabling a nucleic acid architecture that is repeatedly populated during biological strand exchange events.

## Introduction

Although structural and mechanistic information is available for various types of DNA strand exchange processes [1–8], comparatively little is known about RNA/DNA strand exchange. In this reversible process, a strand of RNA hybridizes to one strand of a DNA duplex while displacing the other strand, requiring concomitant disruption of DNA:DNA base pairs and formation of RNA:DNA base pairs. This process occurs most notably at the boundaries of R-loops, such as those left by transcriptional machinery [9], those employed by certain transposons [10, 11], or those created by CRISPR-Cas (clustered regularly interspaced short

**Funding:** J.C.C. is a recipient of the NSF Graduate Research Fellowship (https://www.nsfgrfp.org/). G. J.K was supported by an NHMRC Investigator Grant (EL1, 1175568) (https://www.nhmrc.gov.au/). This work was supported by the Howard Hughes Medical Institute (https://www.hhmi.org/), the National Science Foundation under award number 1817593 (https://www.nsf.gov/), the Centers for Excellence in Genomic Science of the National Institutes of Health under award number RM1HG009490 (https://www.genome.gov/Funded-Programs-Projects/Centers-of-Excellence-in-Genomic-Science), and the Somatic Cell Genome Editing Program of the Common Fund of the National Institutes of Health under award number U01AI142817-02 (https://commonfund.nih.gov/editing). J.A.D. is an HHMI investigator. Beamline 8.3.1 of the Advanced Light Source, a U. S. DOE Office of Science User Facility under Contract No. DE-AC02-05CH11231, is supported in part by the ALS-ENABLE program (http://als-enable.lbl.gov/) funded by the National Institutes of Health, National Institute of General Medical Sciences, grant P30 GM124169-01. The funders had no role in study design, data collection and analysis, decision to publish, or preparation of the manuscript.

**Competing interests:** I have read the journal's policy and the authors of this manuscript have the following competing interests: The Regents of the University of California have patents issued and/or pending for CRISPR technologies on which G.J.K and J.A.D. are inventors. J.A.D. is a cofounder of Caribou Biosciences, Editas Medicine, Scribe Therapeutics, Intellia Therapeutics and Mammoth Biosciences. J.A.D. is a scientific advisor to Caribou Biosciences, Intellia Therapeutics, Scribe Therapeutics, Mammoth Biosciences, Algen Biotechnologies, Felix Biosciences and Inari. J.A.D. is a Director at Johnson & Johnson and at Tempus, and has research projects sponsored by Biogen and Apple Tree Partners. This does not alter our adherence to PLOS ONE policies on sharing data and materials.

palindromic repeats, CRISPR-associated) enzymes during prokaryotic immunity or eukaryotic genome editing [12–15]. Structural insight into RNA/DNA strand exchange could therefore improve our understanding of how transcriptional R-loops are resolved and how CRISPR-Cas enzymes such as Cas9 manipulate R-loops to efficiently reject off-target DNA and recognize on-target DNA.

The defining feature of RNA/DNA strand exchange is the junction where the RNA:DNA helix abuts the DNA:DNA helix. Previous experiments on exchange junctions containing an RNA-5′ end and a DNA-3′ end (an "RNA-5′/DNA-3′ junction," which is the polarity generated by Cas9) showed the component DNA:DNA duplex to be more thermodynamically stable than a free DNA helix end, perhaps due to interhelical RNA:DNA/DNA:DNA stacking [16]. While stacking in DNA-only junctions is thought to occur as it would in an uninterrupted B-form duplex [8, 17, 18], an analogous structural prediction cannot be made for RNA/DNA junctions because the two component helices are predisposed to different geometries: B-form for the DNA:DNA helix and a variant of A-form for the RNA:DNA helix [19–21]. A conformation that preserves base stacking across such a junction must reconcile base pairs that are flat and centered (B-form) with base pairs that are inclined and displaced from the helical axis (A-form). While prior structural studies of Okazaki fragments reckoned with a similar geometric puzzle [22], Okazaki fragments bear an RNA-3′/DNA-5′ polarity (opposite of the polarity addressed here) and lack the strand discontinuity that defines exchange junctions. Thus, the structural basis for the putative stacking-based stability in RNA-5′/DNA-3′ junctions remains unknown.

Here we present the X-ray crystal structure of an RNA-5′/DNA-3′ strand exchange junction, which undergoes an A-to-B transition without loss of base pairing or stacking across the exchange point. This structure reveals the principles of global helical positioning and local adjustments in nucleotide conformation that allow RNA:DNA duplexes to stack on DNA: DNA duplexes in the RNA-5′/DNA-3′ polarity. This model also complements previously determined cryo-electron microscopy structures of DNA-bound Cas9 for which poor local resolution in the original maps prevented accurate modeling of the leading R-loop edge.

## Results

Inspired by previous crystallographic studies of double-stranded DNA dodecamers [23, 24], we designed crystallization constructs that contained a "template" DNA strand (12 nucleotides) and two "exchanging" RNA and DNA oligonucleotides that were complementary to each half of the template DNA strand. In different versions of these constructs, we varied the polarity (RNA-5′/DNA-3′ vs. RNA-3′/DNA-5′) and the internal termini, which were either flush (exchanging oligonucleotides were 6-mers) or extended with a one-nucleotide flap that was not complementary to the template strand (exchanging oligonucleotides were 7-mers, "flapped"). Only the flapped construct in the RNA-5′/DNA-3′ polarity (Fig 1A) yielded well-diffracting crystals (anisotropic resolution of 1.6 to 2.2 Å). Thus, all results discussed here describe a flapped RNA-5′/DNA-3′ strand exchange junction, which is the polarity previously observed to stabilize the component DNA:DNA duplex [16].

We determined the X-ray crystal structure of the exchange junction (Table 1, S1 Fig). In this structure, the asymmetric unit contains three molecules (a "molecule" comprises one DNA 12-mer and its complementary RNA and DNA 7-mers). The crystal lattice is largely stabilized by nucleobase stacking interactions both within and between molecules. Along one lattice direction, Molecules 1 and 2 form a continuous network of stacked helices, in which the external RNA:DNA duplex terminus of each Molecule 1 stacks on the equivalent terminus of Molecule 2, with a similar reciprocal interaction for the external DNA:DNA duplex termini (a

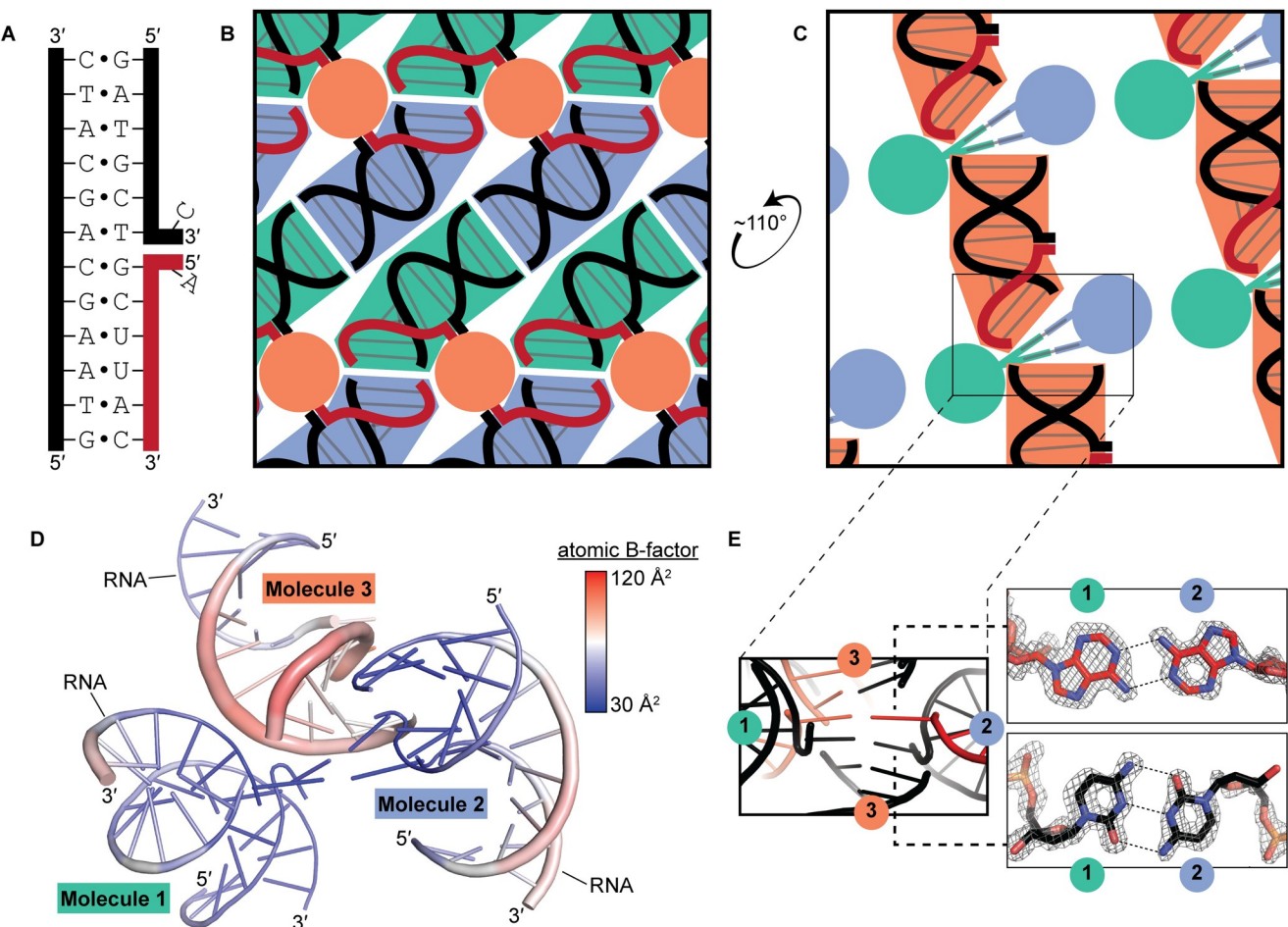

**Fig 1. Stabilizing features of the crystal lattice.** (A) Crystallization construct sequence. Black, DNA; red, RNA. (B) Schematized drawing (not to scale) of the crystal lattice along a direction that depicts the helical network formed by Molecules 1 and 2. Green shading, Molecule 1; blue shading, Molecule 2; orange shading, Molecule 3 (cross section). (C) Similar to panel B, but along a direction that depicts the helical network formed by Molecule 3. (D) Asymmetric unit colored by atomic B-factor. The thickness of the cartoon model also reflects the local B-factors. (E) Model and $2mF_o-DF_c$ map (sharpened by -38 Å$^2$, displayed at 3.3σ) of the Ade-Ade and Cyt-Cyt base pairs (contributed by the flap nucleotides of Molecules 1 and 2) that bridge the helical network formed by Molecule 3. Distortion in the map is due to diffraction anisotropy (see Methods).

"head-to-head" and "foot-to-foot" arrangement) (Fig 1B). Along another lattice direction, symmetry-related instances of Molecule 3 create a head-to-foot helical network (Fig 1C). Compared to Molecules 1 and 2, Molecule 3 is poorly ordered (Fig 1D), and its atomic coordinates appear less constrained by the data due to diffraction anisotropy (see Methods). In the Molecule 3 helical network, two base pairs formed between the flapped nucleotides of Molecules 1 and 2 bridge the duplex ends. The bridging nucleotides form a type I adenine-adenine (ribonucleotide) base pair and a type XV hemiprotonated cytosine-cytosine (deoxyribonucleotide) base pair [25] (Fig 1C and 1E).

The three molecules of the asymmetric unit exhibit canonical Watson-Crick base pairing at all twelve nucleotides of the template DNA strand, and they are generally similar in conformation (RMSD$_{Mol1,Mol2}$ = 0.70 Å; RMSD$_{Mol1,Mol3}$ = 1.5 Å, RMSD$_{Mol2,Mol3}$ = 1.8 Å) (Fig 2A). The most dramatic differences are between Molecules 1/2 and Molecule 3. For example, Molecule 3's flapped nucleotides form no intermolecular base pairs, and the conformation of the DNA flap is flipped relative to Molecules 1/2. Additionally, the external three base pairs of Molecule 3's DNA:DNA helix tilt slightly toward the major groove as compared to the equivalent

**Table 1. Crystallographic data and refinement statistics.**

| | RNA-5′/DNA-3′ strand exchange junction (PDB 7THB) |
|---|---|
| **Data collection** | |
| Wavelength (Å) | 1.116 |
| Resolution range (Å) | 35.3–1.64 (1.78–1.64) |
| Diffraction limit #1 (Å) | 1.66 |
| Principal axes (orthogonal basis) | 0.865, -0.0396, -0.501 |
| Principal axes (reciprocal lattice) | 0.657 a* - 0.168 b* - 0.735 c* |
| Diffraction limit #2 (Å) | 2.18 |
| Principal axes (orthogonal basis) | 0.168, 0.962, 0.213 |
| Principal axes (reciprocal lattice) | 0.152 a* + 0.981 b* + 0.117 c* |
| Diffraction limit #3 (Å) | 1.64 |
| Principal axes (orthogonal basis) | 0.473, -0.269, 0.839 |
| Principal axes (reciprocal lattice) | 0.397 a* - 0.342 b* + 0.852 c* |
| Space group | P 1 |
| Unit cell | |
| a, b, c (Å) | 37.0, 43.6, 52.2 |
| α, β, γ (°) | 92.1, 103.7, 100.0 |
| Total reflections | 147975 (7830) |
| Unique reflections | 24808 (1240) |
| Multiplicity | 6.0 (6.3) |
| Spherical completeness (%) | |
| 35.3–1.64 Å | 64.7 |
| 35.3–2.22 Å | 97.2 |
| 1.78–1.64 Å | 14.5 |
| Ellipsoidal completeness (%) | |
| 35.3–1.64 Å | 86.8 |
| 35.3–2.22 Å | equivalent to spherical completeness, by definition |
| 1.78–1.64 Å | 50.0 |
| $<I/\sigma(I)>$ | 15.6 (1.5) |
| Wilson B-factor (Å$^2$) | |
| Eigenvalue #1 (Å) | 48.6 |
| Principal axes (orthogonal basis) | 0.960, -0.166, -0.224 |
| Principal axes (reciprocal lattice) | 0.799 a* - 0.323 b* - 0.507 c* |
| Eigenvalue #2 (Å) | 86.7 |
| Principal axes (orthogonal basis) | 0.226, 0.935, 0.275 |
| Principal axes (reciprocal lattice) | 0.209 a* + 0.961 b* + 0.183 c* |
| Eigenvalue #3 (Å) | 45.5 |
| Principal axes (orthogonal basis) | 0.164, -0.315, 0.935 |
| Principal axes (reciprocal lattice) | 0.123 a* - 0.300 b* + 0.946 c* |
| $R_{merge}$ | 0.037 (1.293) |
| $R_{meas}$ | 0.041 (1.410) |
| $R_{pim}$ | 0.016 (0.556) |
| $CC_{1/2}$ | 0.999 (0.474) |
| **Refinement** | |
| Resolution range (Å) | 35.3–1.64 (1.77–1.64) |
| Reflections used in refinement | 24717 (1054) |
| Reflections used for $R_{free}$ | 1223 (41) |

*(Continued)*

**Table 1.** (Continued)

| | RNA-5′/DNA-3′ strand exchange junction (PDB 7THB) |
|---|---|
| $R_{work}$ | 0.237 (0.369) |
| $R_{free}$ | 0.284 (0.356) |
| $CC_{work}$ | 0.912 (0.616) |
| $CC_{free}$ | 0.939 (0.562) |
| Number of non-hydrogen atoms | 1651 |
| macromolecules | 1584 |
| ligands | 0 |
| solvent | 67 |
| Protein residues | 0 |
| RMSD–bond lengths (Å) | 0.014 |
| RMSD–angles (°) | 1.42 |
| Coordinate error (maximum-likelihood based estimate) (Å) | 0.30 |
| Clashscore | 0.00 |
| Average B-factor | 59.8 |
| macromolecules | 60.2 |
| solvent | 50.4 |
| Number of TLS groups | 15 |

Diffraction limits and eigenvalues of overall anisotropy tensor on |*F*|s are displayed alongside the corresponding principal axes of the ellipsoid fitted to the diffraction cut-off surface as direction cosines in the orthogonal basis (standard PDB convention), and in terms of reciprocal unit-cell vectors. Statistics for the highest-resolution shell are shown in parentheses.

positions of Molecules 1/2. Notably, the similarity of all three molecules at the three base pairs on either side of the exchange point ($RMSD_{Mol1,Mol2}$ = 0.57 Å; $RMSD_{Mol1,Mol3}$ = 0.50 Å, $RMSD_{Mol2,Mol3}$ = 0.75 Å) suggests that the conformation in this region represents a low-energy solution to the stacking of RNA:DNA and DNA:DNA helices.

At the exchange point of Molecules 1 and 2, the flapped nucleotides are stabilized not only by intermolecular base pairing (Fig 1C and 1E) and intramolecular stacking (Fig 2B), but also by hydrogen bonds between sugar hydroxyls and backbone phosphates. Specifically, at the junction-proximal phosphodiester within the DNA:DNA helix, the *pro-S*$_p$ and *pro-R*$_p$ oxygens are hydrogen-bonded to the terminal 3′ hydroxyl of the flapped DNA nucleotide and the terminal 5′ hydroxyl of the flapped RNA nucleotide, respectively. Additionally, the *pro-S*$_p$ oxygen of the flapped DNA nucleotide is hydrogen-bonded to the 2′ hydroxyl of the flapped RNA nucleotide (Fig 2B). If the flaps were longer than one nucleotide, as would occur during biological strand exchange events, the hydrogen bonds to the terminal 3′/5′ hydroxyls would be perturbed. However, in Molecule 3, the flipped deoxycytidine conformation precludes all the mentioned extrahelical hydrogen bonds, yet the base-paired nucleotides within the junction are conformationally similar to the same region in Molecules 1 and 2 (Fig 2A). Therefore, we expect that the structural features of interest to this work—that is, the conformation of the base-paired nucleotides immediately adjacent to the junction—would be populated by junctions bearing flush RNA/DNA ends or flaps of arbitrary length. On the other hand, the flap conformations and the intermolecular base pairs observed here are peculiarities of the crystal lattice. During biological strand exchange processes, these overhung nucleotides would be unpaired and disordered [8].

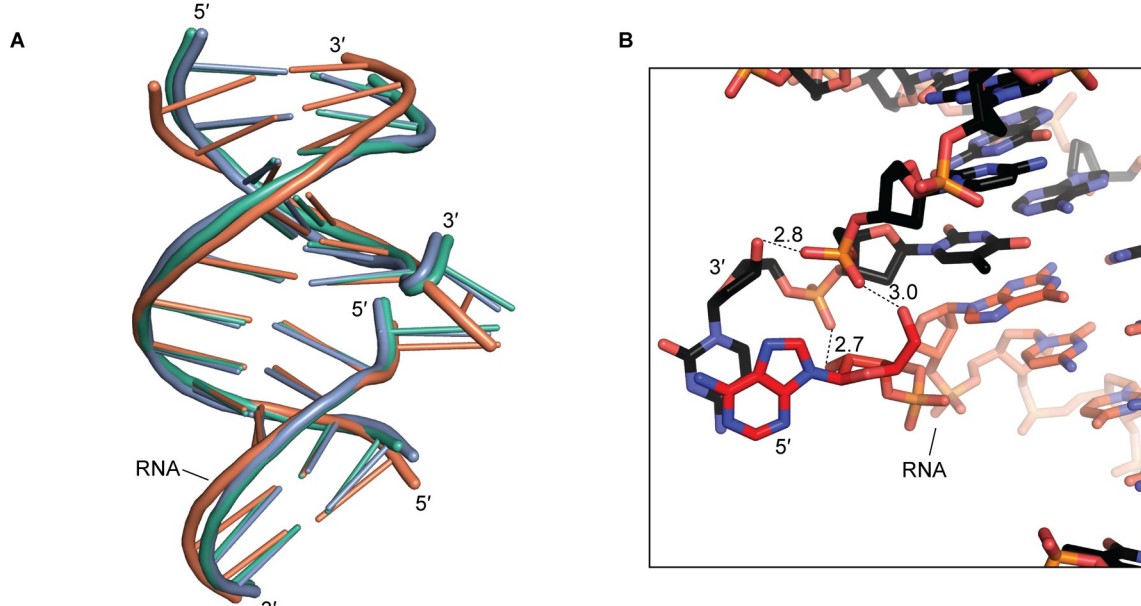

**Fig 2. Molecule-to-molecule similarity and hydrogen bonding at the flapped nucleotides.** (A) All-atom alignment of the three molecules in the asymmetric unit. Green, Molecule 1; blue, Molecule 2; orange, Molecule 3. Molecules 2 and 3 were aligned to Molecule 1 in this depiction. (B) Hydrogen bonding at the flapped nucleotides of Molecule 1. Dotted lines indicate hydrogen bonds, and adjacent numbers indicate interatomic distance in Å. Black, DNA; red, RNA. This hydrogen bonding pattern is also observed in Molecule 2 but not in Molecule 3.

To understand the nature of the transition in helical geometry across the junction, we performed alignments of regularized A-form and B-form DNA:DNA helices with the observed RNA:DNA and DNA:DNA helices, respectively. These alignments revealed that the DNA:DNA helix closely approximates perfect B-form geometry, especially in the nucleotides closest to the junction (Fig 3A–3C). Likewise, the RNA strand of the RNA:DNA helix closely approximates A-form geometry (Fig 3A–3C). On the other hand, the DNA strand of the RNA:DNA helix deviates from its A-form trajectory in the three nucleotides that approach the exchange point, where the backbone is compressed toward the minor groove (Fig 3B and 3D).

Interestingly, calculation of $z_P$, a geometric parameter that differentiates A-form from B-form base steps [26], indicated that the RNA:DNA base step adjacent to the exchange point is A-like, while the base steps in the center of the RNA:DNA helix are intermediate in their A/B character (Fig 4A). This result indicates an important distinction between strand trajectory (in terms of global alignment to a regularized A-form or B-form helix) and the local nucleotide conformations that underlie the trajectory. In the RNA:DNA helix, the departure from A-form trajectory observed at junction-adjacent nucleotides appears to result from non-A conformations at more junction-distal nucleotides. Other indicators of helical geometry also suggest a mixture of A and B character across the RNA:DNA helix (S2 Fig).

To probe helical geometry with strand specificity, we calculated χ and δ, nucleotide torsion angles that differ in A-form vs. B-form helices [27]. These parameters revealed that the irregularities observed in the paired base step parameters (Fig 4A and S2 Fig) arise entirely from the template DNA strand, which flips between A- and B-like conformations within the RNA:DNA hybrid (Fig 4B and S3 Fig). In contrast, the RNA strand is entirely A-like, and all nucleotides of the DNA:DNA helix are B-like except at position 12 of the continuous strand, which is likely due to an end effect. These observations agree with the conclusions drawn from the alignments

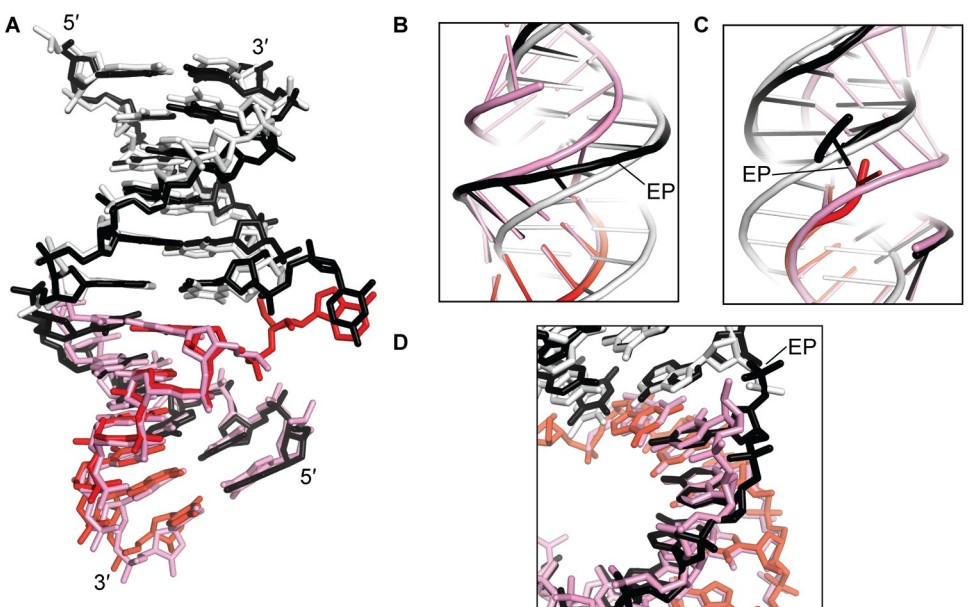

**Fig 3. Alignments to regularized A-form/B-form helices.** (A) Black, DNA of Molecule 1; red, RNA of Molecule 1; white, regularized B-form DNA:DNA helix aligned to the 6 bp of Molecule 1's DNA:DNA helix; pink, regularized A-form DNA:DNA helix aligned to the 6 bp of Molecule 1's RNA:DNA helix. (B) Cartoon depiction, focused on the continuous strand. The alignment procedure for each 6-bp block was identical to that performed in panel A, but in this depiction, the B-form (white) and A-form (pink) helices were extended by an additional 6 bp (extended nucleotides were not considered during alignment) to illustrate the path that the helix would take if continuing along a perfect B-form or A-form trajectory. EP, exchange point (that is, the phosphodiester or gap lying between the two nucleotides where the helix changes from RNA:DNA to DNA:DNA). (C) Similar to panel B, but focused on the discontinuous strand. (D) Close-up of the same representation depicted in panel A, focused on the nucleotides that deviate most dramatically from the aligned A-form helix.

(Fig 3A), and they highlight the DNA strand of the RNA:DNA helix as the structure's most geometrically irregular region, which may enable the junction-adjacent deviation in trajectory.

In addition to the distortions in the continuous DNA strand, the geometric switch also seems to depend on the break in the discontinuous strand, which facilitates a marked jump in the backbone trajectory across the exchange point (Fig 3C). This feature reflects a global jump in helical positioning that is visualized most clearly in the aligned regularized A-form and B-form duplexes, whose helical axes are tilted and misaligned with respect to each other (the helical axes are tilted from parallel by 14˚, Mol1; 18˚, Mol2; 2˚, Mol3) (Figs 2A, 3B and 3C). Axis misalignment is detectable in the large positive y-displacement value across the central base step, which deviates dramatically from the expected value (0 Å) for either an A-form or B-form duplex (Fig 4C). This observation emphasizes the exchange point as a special base step with noncanonical alignment, made possible by discontinuity in the exchanging strands.

## Discussion

Together, our data suggest that stacking an RNA:DNA helix on a DNA:DNA helix does not require deviation of the RNA strand or either strand of the DNA:DNA helix from their native A-form or B-form conformations, respectively. Instead, continuous stacking appears to result from a combination of three structural principles. First, alternating A-like and B-like nucleotide conformations in the hybrid's DNA strand compress the strand relative to a pure A-form trajectory (Figs 3B, 3D, 4B and 5A). Due to A-form base pair inclination (~20˚ from perpendicular to the helical axis) in RNA:DNA duplexes, the DNA naturally juts further along the

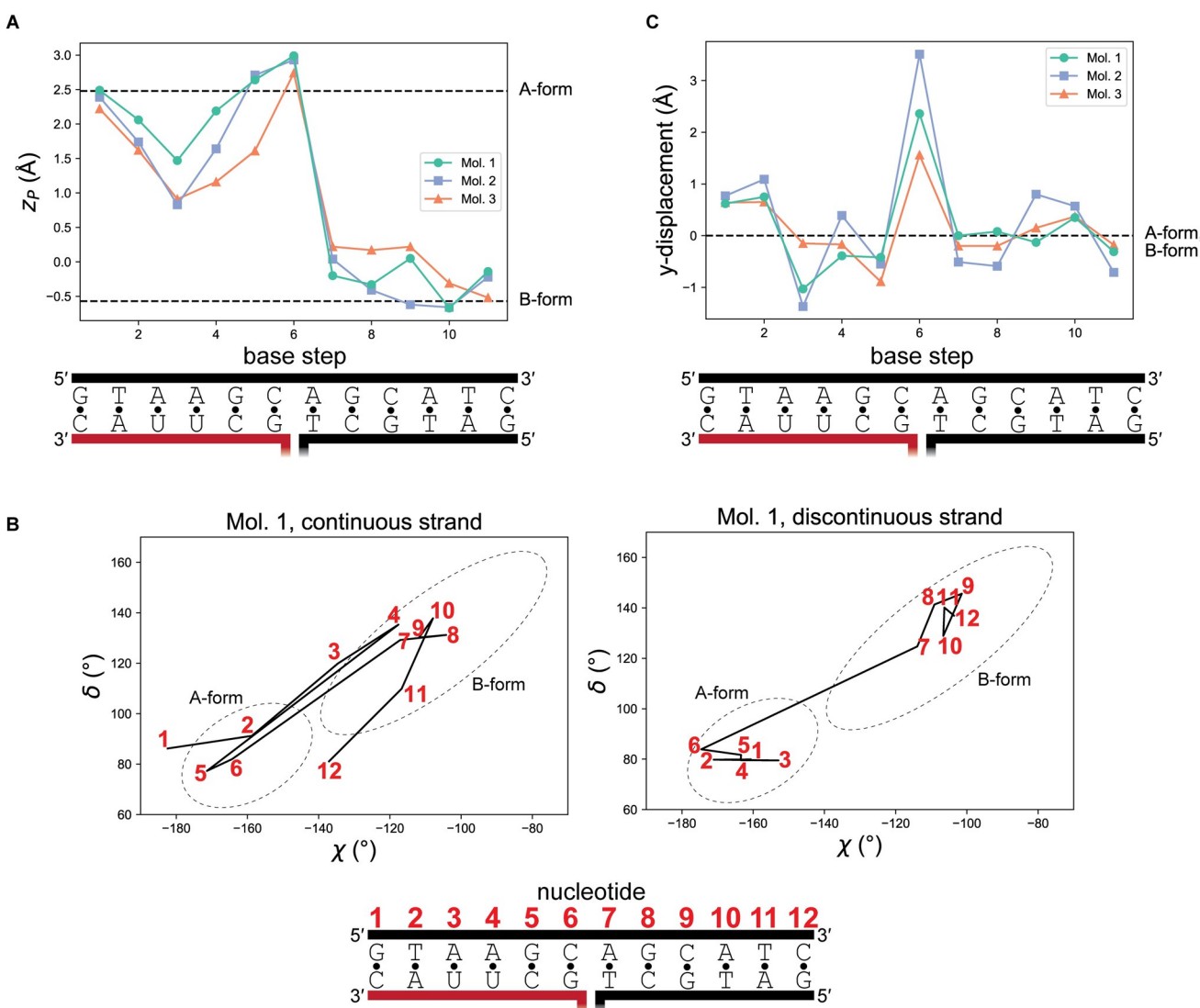

**Fig 4. Geometric details of the A-to-B transition.** (A) For a given base step, the parameter $z_P$ is the mean of the $z$-displacement of the two phosphorus atoms from the dimer's reference $xy$-plane. Note that $z_P$ is defined by a pair of dinucleotides, so there are only 11 data points for a 12-bp helix, and integral x-values lie between the base pairs in the diagram. This parameter was originally introduced for its utility in distinguishing A-form from B-form base steps. Black, DNA; red, RNA. (B) $\chi$ and $\delta$ are the two nucleotide torsion angles that best distinguish A-form from B-form geometry. Note that these torsion angles are defined for each individual nucleotide, so there are 24 data points for a 12-bp helix. Integers in red refer to individual nucleotides, as indicated in the schematic at the bottom. Dashed ellipses were drawn to match those depicted in [27]. (C) Y-displacement. Similar to $z_P$, this parameter describes base steps (pairs of dinucleotides), not individual nucleotides. This parameter cannot distinguish A-form from B-form geometry. Instead, note that the base step across the exchange point dramatically departs from both A-form and B-form geometry.

helical axis than the RNA at the RNA-5′ end. This slanted RNA:DNA end can be stacked upon a flat DNA:DNA end through strand-specific compression—that is, compression of the hybrid's protruding DNA strand (Fig 5A). Second, an alternative to strand compression is to tilt the helical axes themselves, which occurs in Molecules 1 and 2 but not Molecule 3 (Figs 2A and 5A). Third, the helical centers are misaligned at the exchange point (Figs 3B, 3C and 4C), which effectively aligns the off-center base pairs of the A-form duplex with the centered base pairs of the B-form duplex (Fig 5B).

This new structure is best examined in the context of previous structural studies of RNA: DNA/DNA:DNA junctions emulating Okazaki fragments, which include a chimeric

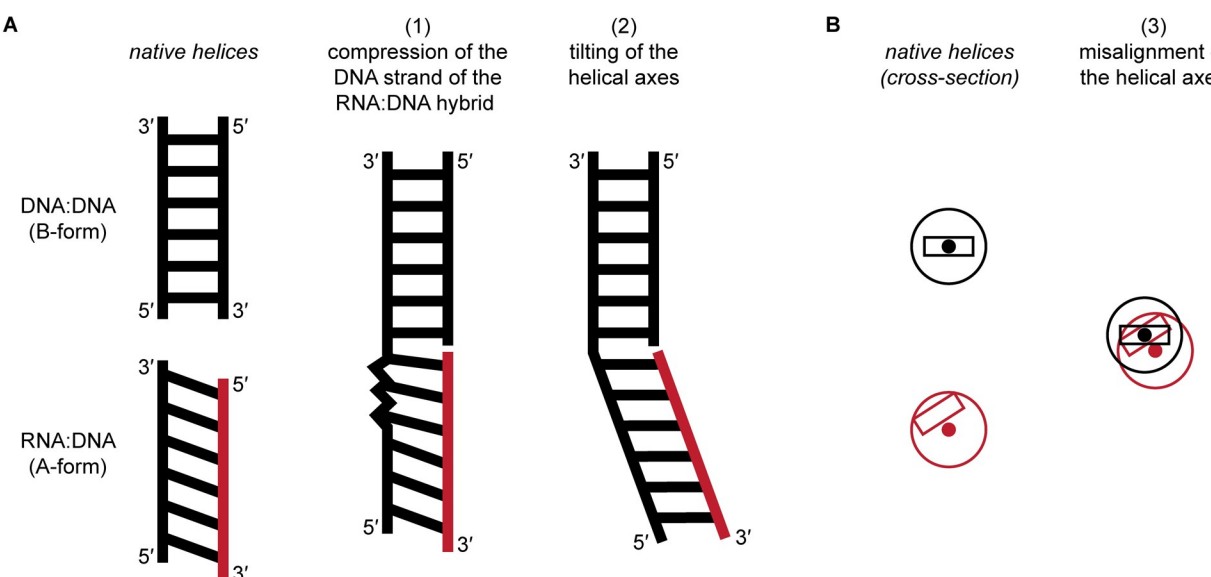

**Fig 5. Structural principles of A-on-B stacking at the RNA-5′/DNA-3′ strand exchange junction.** (A) Simplified schematics illustrating strand-specific compression and tilting of the helical axes. The slanted appearance of the RNA:DNA duplex is intended to represent the base pair inclination characteristic of A-form duplexes, which pushes the 3′ DNA end farther along the helical axis than the 5′ RNA end. Black, DNA; red, RNA. (B) Helical cross-sections. Black, DNA:DNA helix; red, RNA:DNA helix. The rectangle represents the base pair nearest the exchange point (centered in the B-form helix, off-center in the A-form helix). The solid circle represents the helical axis. The true stacking solution is a combination of the three principles illustrated here, although Molecule 3 does not exhibit tilting.

(covalently continuous) RNA-DNA strand. When crystallized, these fragments assumed an entirely A-form conformation, even within the DNA:DNA duplex [28–32]. However, in solution, Okazaki fragments resembled the present structure in that they were A-like within the RNA:DNA helix and B-like within the DNA:DNA helix [22, 33–36]. Solution structures also exhibited a tilt between the RNA:DNA/DNA:DNA helical axes and intermediate nucleotide geometry within the DNA of the hybrid. Because intermediate geometry is a known feature of the DNA of any RNA:DNA hybrid [19, 20], it may be the natural inclination of this more geometrically ambiguous strand to accommodate the A-to-B transition as it does in the present structure. Notably, dramatic misalignment of the RNA:DNA/DNA:DNA helical centers is observed only in the present structure and is likely enabled by the break in the exchanging strands, which is not a feature of Okazaki fragments.

Because stable stacking of another duplex on a DNA:DNA terminus is expected to inhibit duplex melting [37], the structural principles illuminated here may explain the rigidity that we previously observed in the DNA:DNA duplex of RNA-5′/DNA-3′ exchange junctions [16]. However, it is also possible that different sequences or environments promote different conformational preferences than those observed in this crystal structure. Previously, we also observed that the DNA:DNA duplex in junctions of the opposite polarity (RNA-3′/DNA-5′) is destabilized relative to a non-exchanging terminus [16]. Unfortunately, because that junction type failed to crystallize under our tested conditions, this odd asymmetry in junction structure remains unexplained.

Nevertheless, the stacked RNA-5′/DNA-3′ structure determined here represents a key conformation that is likely populated throughout RNA/DNA exchange events, including those mediated by the genome-editing protein Cas9. Branch migration is crucial to Cas9 target search, which involves repeated R-loop formation (RNA invades a DNA:DNA duplex) and resolution (DNA invades an RNA:DNA duplex) until the true target is located [15]. During this process, the leading R-loop edge likely passes through interhelically stacked states between

base pair formation and breakage events. Consistent with this prediction, in some cryo-electron microscopy structures depicting Cas9-bound R-loops, the leading (RNA-5′/DNA-3′) R-loop edge appeared interhelically stacked [38, 39]. While local resolution was insufficient to enable accurate atomic modeling of the exchange junction from the original electron microscopy maps, our high-resolution crystal structure provides a new geometric standard for modeling this kind of junction.

Importantly, exchange junctions are dynamic structures, and each time an R-loop grows or shrinks, stacking must be disrupted at the junction [8]. Thus, in addition to the stacked structure determined here, which can be interpreted as a ground state, strand exchange also requires passage through unstacked conformations, some of which may resemble the junction structures seen in other Cas9-bound R-loops [40, 41]. A complete model of RNA/DNA strand exchange, then, will rely on a structural and energetic understanding of the junction in both stacked and unstacked states, and it will account for the effects of the proteins acting in R-loop formation and resolution.

## Methods

### Oligonucleotide synthesis and sample preparation

All oligonucleotides (DNA 12-mer {5′−GTAAGCAGCATC−3′}; DNA 7-mer {5′−GATGCTC−3′}; RNA 7-mer {5′−AGCUUAC−3′}) were synthesized and purified by Integrated DNA Technologies (high-performance liquid chromatography (HPLC) purification for DNA oligonucleotides and RNase-free HPLC purification for the RNA oligonucleotide). Dry oligonucleotides were dissolved in nuclease-free water (Qiagen), and concentrations were estimated by Nanodrop (Thermo Scientific) absorbance measurements with extinction coefficients estimated according to [42] (DNA 12-mer, $\varepsilon_{260}$ = 135200 $M^{-1} \cdot cm^{-1}$; DNA 7-mer, $\varepsilon_{260}$ = 70740 $M^{-1} \cdot cm^{-1}$; RNA 7-mer, $\varepsilon_{260}$ = 75580 $M^{-1} \cdot cm^{-1}$). The three oligonucleotides were combined and diluted in water, each at 500 μM final concentration. This exchange junction sample was incubated at 50°C for 10 minutes, cooled to 25°C within a few seconds, and used directly in the crystallization setups described below.

### Crystallization and data collection

Initial screens were performed using Nucleix and Protein Complex suites (Qiagen) in a sitting-drop setup, with 200 nL of sample added to 200 nL of reservoir solution by a Mosquito instrument (SPT Labtech) and incubated at either 4°C or 20°C. Several conditions yielded crystals within one day, and initial hits were further optimized at a larger scale. The crystal used for the final dataset was produced as follows: 0.5 μL of sample was combined with 0.5 μL reservoir solution (0.05 M sodium succinate (pH 5.3), 0.5 mM spermine, 20 mM magnesium chloride, 2.6 M ammonium sulfate) in a hanging-drop setup over 500 μL reservoir solution, and the tray was stored at 20°C. Crystals formed within one day and remained stable for the 2.5 weeks between tray setting and crystal freezing. A crystal was looped, submerged in cryo-protection solution (0.05 M sodium succinate (pH 5.3), 0.5 mM spermine, 20 mM magnesium chloride, 3 M ammonium sulfate) for a few seconds, and frozen in liquid nitrogen. Diffraction data were collected under cryogenic conditions at the Advanced Light Source beamline 8.3.1 on a Pilatus3 S 6M (Dectris) detector.

### Data processing, phase determination, and model refinement

Preliminary processing of diffraction images was performed in XDS [43, 44]. Unmerged reflections underwent anisotropic truncation, merging, and anisotropic correction using the default

parameters of the STARANISO server (v3.339) [45], and a preliminary structural model was included in the input to estimate the expected intensity profile. The best-fit cut-off ellipsoid imposed diffraction limits of 1.66 Å, 2.18 Å, and 1.64 Å based on a cut-off criterion of I/σ(I) = 1.2. The "aniso-merged" output MTZ file was used for downstream processing. Using programs within CCP4 (v7.1.015), $R_{free}$ flags were added to 5% of the reflections, and reflections outside the diffraction cut-off surface were removed.

Phases were determined by molecular replacement with Phaser-MR [46], as implemented in Phenix v1.19.2–4158 [47]. The search model comprised two components (unconstrained with respect to each other), both generated in X3DNA v2.4 [48] and each representing one half of the base-paired portion of the crystallization construct. The first component was a 6-base-pair RNA:DNA duplex with perfect A-form geometry and sequence 5′-GCUUAC-3′ / 5′-GTAAGC-3′ (created using the program "fiber" with the -rna option, followed by manual alteration of the DNA strand in PyMOL v2.4.1). The second component was a 6-base-pair DNA:DNA duplex with perfect B-form geometry and sequence 5′-GATGCT-3′ / 5′-AGCATC-3′ (created with "fiber" option -4). Successful phasing was achieved by searching for three copies of each of these components (six components total). Additional phosphodiesters and nucleotides were built in Coot v0.9.2 [49], and the model underwent iterative refinements in Phenix. Phasing and preliminary refinements were initially performed using an earlier (lower-resolution) dataset that had similar unit cell parameters to the final dataset described above.

The initial model, which was refined into a map generated from the earlier dataset, was rigid-body docked into the final-dataset-derived map and underwent further iterative refinements, beginning with resetting of the atomic B-factors, simulated annealing, and addition of ordered solvent. Non-crystallographic symmetry restraints were applied in early rounds of refinement to link the torsion angles of the three molecules within the asymmetric unit; these restraints were removed in the final rounds of refinement. TLSMD [50, 51] was used to determine optimal segmentation for Translation/Libration/Screw (TLS) refinement (each 7-mer comprised a separate segment, and the 12-mers were each divided into three segments: nucleotides 1–4, 5–8, 9–12). Refinement using Phenix's default geometry library yielded dozens of bond lengths and angles that were marked as outliers by the PDB validation server, so the faulty parameters were rigidified *ad hoc* (that is, their estimated standard deviation values in the library files were made smaller, with no change to the mean values). The final three cycles of refinement were performed in Phenix with adjustments to XYZ (reciprocal-space), TLS (segments as indicated above), and individual B-factors. In Table 1, STARANISO and Phenix were used to calculate the data collection statistics and the refinement statistics, respectively. The composite omit map displayed in S1 Fig was generated by Phenix's CompositeOmit job ("anneal" method; 5% of atoms omitted in each group; missing $F_{obs}$ left unfilled; $R_{free}$-flagged reflections included).

The final $R_{free}$ value (0.284) is higher than expected for a structure refined using diffraction data at a resolution of 1.6 Å [52]. However, it is important to note that the highest-resolution shell has a completeness of just 6%, and completeness only rises above 95% at ~2.3 Å, due mostly to the anisotropic nature of the diffraction data. Additionally, due to diffraction anisotropy, the $2mF_o$-$DF_c$ map appears distorted along certain dimensions, affecting interpretation of Molecule 3 most negatively. Therefore, the geometric details of Molecule 3's phosphate backbone are poorly constrained, and Molecule 1 or 2 should instead be considered as the most accurate representation of the structure. Anisotropy also prevented identification of water molecules around Molecule 3. Furthermore, the $mF_o$-$DF_c$ map revealed several globular patches of positive density in the major and minor grooves of all molecules, 3.5–4 Å away from the nearest nucleic acid atom. Because these patches bore no recognizable geometric

features, attempts to model them with buffer components failed to improve $R_{free}$, so they were left unmodeled. Any of the mentioned issues may contribute to the high $R_{free}$ value.

Beyond the anisotropy, the overall high B-factors in this structure produce $2mF_o-DF_c$ density that is "blurred" (S1 Fig) [53]. To enhance high-resolution features of the map for visual inspection and figure preparation, Coot's Map Sharpening tool was used. B-factor adjustments used for sharpening are reported in the figure legend. Sharpening only effectively revealed high-resolution features for Molecule 1 or 2, as density from Molecule 3 is too anisotropically distorted.

## Structure analysis and figure preparation

Structural model and map figures were prepared in PyMOL. Alignments were performed using PyMOL's "align" function without outlier rejection. Regularized A-form and B-form DNA:DNA duplexes were prepared using X3DNA's "fiber" program (options -1 and -4, respectively), using the same sequence present in the helical portion of the crystallization construct (except RNA was modeled as the corresponding DNA sequence). While the A-form DNA:DNA helix may not perfectly represent a regularized version of the RNA:DNA helix with our sequence [19, 20], "fiber" does not permit generation of RNA:DNA helices with generic sequence, and the general geometric features of A-form DNA:DNA vs. A-form RNA:DNA are expected to be similar enough to support the conclusions drawn in this work. Base step and nucleotide geometric parameters were calculated using the "find_pair" and "analyze" programs within X3DNA. On graphs of these parameters, dashed lines indicating the expected value for A-form or B-form DNA were calculated by performing an equivalent analysis on the X3DNA-generated regularized A-form/B-form helices and taking the average across all base steps/nucleotides, unless indicated otherwise. Nucleotides with A/B character exhibit a spread of values around those indicated by the dashed lines (as represented more accurately by the dashed ellipses in Fig 4B), and the dashed lines are drawn merely to guide the reader's eye to general trends. Angles between the helical axes of the DNA:DNA and RNA:DNA duplex were calculated as the angle between the helical axis vectors of the aligned regularized A-form and B-form helices. Graphs were prepared using matplotlib v3.3.2 [54]. Final figures were prepared in Adobe Illustrator v25.4.1.

## Supporting information

**S1 Fig. Overview of the asymmetric unit.** Model and composite omit $2mF_o-DF_c$ map (displayed at 1.5σ) of the asymmetric unit. Black, DNA; red, RNA. For clarity, the displayed density is truncated 2 Å from the atoms displayed in the model. "Blurriness" of the electron density is due to high atomic B-factors [53].
(TIF)

**S2 Fig. Additional geometric details of the A-to-B transition.** (A) X-displacement of the 11 base steps of the 12-bp helix. Black, DNA; red, RNA. (B) Inclination of the 11 base steps of the 12-bp helix. (C) Slide of the 11 base steps of the 12-bp helix. (D) Pseudorotation phase angles for the ribose/deoxyribose conformation at every nucleotide within the 12-bp helix (24 data points per molecule). The modeled sugar conformations might not be unique solutions for this dataset, as in many cases these structural details cannot be directly discerned from the $2mF_o-DF_c$ map. For this dataset, the most reliable parameters are those defined directly by the nucleobase and phosphate positions, which appear clearly in the $2mF_o-DF_c$ map (and likely impose indirect geometric constraints on the sugar pucker).
(TIF)

**S3 Fig. Nucleotide torsion angles for Molecules 2 and 3.** Analogous to Fig 4B.
(TIF)

## Acknowledgments

We thank J.M. Holton and J.H. Cate for data processing advice. We thank G. Meigs for technical assistance at the beamline. We thank J. Kuriyan for scientific advice.

## Author Contributions

**Conceptualization:** Joshua C. Cofsky, Jennifer A. Doudna.

**Formal analysis:** Joshua C. Cofsky, Gavin J. Knott, Christine L. Gee.

**Funding acquisition:** Jennifer A. Doudna.

**Investigation:** Joshua C. Cofsky, Gavin J. Knott.

**Supervision:** Jennifer A. Doudna.

**Visualization:** Joshua C. Cofsky.

**Writing – original draft:** Joshua C. Cofsky.

**Writing – review & editing:** Joshua C. Cofsky, Gavin J. Knott, Christine L. Gee, Jennifer A. Doudna.

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
