## [Decision Letter · Decision Letter 0]

18 Feb 2022

PONE-D-22-01714Crystal structure of an RNA/DNA strand exchange junctionPLOS ONE

Dear Dr. Doudna,

Thank you for submitting your manuscript to PLOS ONE. After careful consideration, we feel that it has merit but does not fully meet PLOS ONE’s publication criteria as it currently stands. Therefore, we invite you to submit a revised version of the manuscript that addresses the points raised during the review process. As you see, all 3 reviewers appreciate the quality of the work and the manuscript. They all have minor comments, which I would like you to address in the revised version, either in the form of comments on the feedback, changes in the manuscript as suggested, or as supplementary files/figures. 

We look forward to receiving your revised manuscript.

Kind regards,

Petri Kursula

Academic Editor

PLOS ONE

Journal Requirements:

"I have read the journal's policy and the authors of this manuscript have the following competing interests: The Regents of the University of California have patents issued and/or pending for CRISPR technologies on which G.J.K and J.A.D. are inventors. J.A.D. is a cofounder of Caribou Biosciences, Editas Medicine, Scribe Therapeutics, Intellia Therapeutics and Mammoth Biosciences. J.A.D. is a scientific advisor to Caribou Biosciences, Intellia Therapeutics, Scribe Therapeutics, Mammoth Biosciences, Algen Biotechnologies, Felix Biosciences and Inari. J.A.D. is a Director at Johnson & Johnson and at Tempus, and has research projects sponsored by Biogen and Apple Tree Partners."

Reviewers' comments:

Reviewer's Responses to Questions

**Comments to the Author**

1. Is the manuscript technically sound, and do the data support the conclusions?

Reviewer #1: Yes

Reviewer #2: Yes

Reviewer #3: Yes

2. Has the statistical analysis been performed appropriately and rigorously? 

Reviewer #1: Yes

Reviewer #2: Yes

Reviewer #3: Yes

3. Have the authors made all data underlying the findings in their manuscript fully available?

Reviewer #1: Yes

Reviewer #2: Yes

Reviewer #3: Yes

4. Is the manuscript presented in an intelligible fashion and written in standard English?

Reviewer #1: Yes

Reviewer #2: Yes

Reviewer #3: Yes

5. Review Comments to the Author

Reviewer #1: The paper describes a crystal structure in which a DNA oligomer makes complementary interactions with two shorter duplexes: one DNA and one RNA. Structures in which one of the complementary strands of a double DNA helix is replaced in part by RNA occur in some important biological processes. The question, from the structural point of view, is how the DNA, with its propensity to form B-type helices, accommodates RNA, which usually forms very different A-type helices. Of special interest in this case is the junction where the DNA-DNA duplex turns into DNA-RNA.

The science is sound, meticulously performed, the data have been carefully processed, the attached validation report is very neat. The paper is well written and illustrated. The results are presented concisely, the discussion is well balanced.

No PDB code is quoted in the paper but it is in the validation report and I have verified that the structural model is held waiting for publication.

A minor point: some numbers are quoted to more decimal places than is probably reasonable. E.g. resolution range 35.335-1.637. Or RMSD values quoted to thousandths of an Angstrom. It's up to us to assess the accuracy of the values printed out by the software.

Reviewer #2: In the manuscript Cofsky and colleagues decribe a crystal structure of RNA/DNA complex that mimics a strand exchange function.The study is carefully carried out and structure is analyzed to fine detail. Descriptions on the methods are clear and authors disclose the problems also faced when interpreting the diffraction data, during reciprocal refinement and when interpreting the electron density maps affected by the anisotropy of the data. The coordinates, structure factors and raw images have been made available to the community (on hold for publication) and this is important especially in these problematic cases as it allows improvement of the software used in the field. Overall the analysis and writing is very careful and I have only minor suggestions to the authors how to improve the manuscript for the publication.

The data is very anisotropic and the authors describe this very clearly in the manuscript. I would however suggest that the authors would add a third resolution range to the data table which clearly indicates that the completeness is better than currently the numbers imply. This would, based on the description in the Methods section be [35.335 - 2.3 Å]. There is a slight discrepancy, as one would expect the ellipsoidal completeness to be higher than in the current table 1.

The authors use sharpened electron density maps in figure 1E (unsharpened comparison could be added in the supplement). However they describe well in the methods section that the density is not very clear for the different molecules. As careful analysis describing different variations in the A and B form helices is carried out, it would be good to add some representative overall electron density maps to the supplementary figures. As also the rmsd values were adjusted in the refinement to restrain the geometries this would allow better transparency for the reader.

The high R-factor is slightly surprising for the resolution despite the anisotropy as authors also disclose. TLS refinement was carried out so analysis of the 15 TLS groups would be beneficial and could provide some insight to the molecule 3 disorder. The selected TLS segments are quite small. It would be slightly suprising if 7-mers and dissected pieces of 12-mers would not "move" in concert.

It is not entirely evident, when analyzing the structures, which features are resulting from crystal packing and which are expected interactions in a biological system. Solution data could be helpful, but authors could at least sharpen this discussion when discussing about intermolecular base pairing.

Reviewer #3: Cofsky and co-workers present the crystal structure of a RNA/DNA hybrid strand exchange junction. Their structure sheds light on the geometrical properties of a RNA/DNA strand exchange junction showing a mixture of A-and B-form. This is a well described work that has been conducted thoroughly. I appreciate the illustration of the crystal lattice parameters and how the different molecules in the AU compare to each other in orientation and similarity. They go on with a detailed description of the geometric parameters resulting in some governing principles. They argue that their structure represents a key conformation especially for Cas9 genome editing, that can be useful in better modelling of cryo EM maps that have a much lower resolution.

I am genuinely happy with the presented results and the thoroughness of the analysis. However, I would appreciate more statistics on the model data like mean coordinate error and also an overall representation of the electron density fit. This would help to assess how accurate the model is in addition to the wealth of statistics provided by the authors.

6. PLOS authors have the option to publish the peer review history of their article (what does this mean?). If published, this will include your full peer review and any attached files.

Reviewer #1: No

Reviewer #2: No

Reviewer #3: No

---

## [Author Response · Author response to Decision Letter 0]

2 Mar 2022

See "Response to Reviewers" document

---

## [Editor Report · Decision Letter 1]

7 Mar 2022

Crystal structure of an RNA/DNA strand exchange junction

PONE-D-22-01714R1

Dear Dr. Doudna,

We’re pleased to inform you that your manuscript has been judged scientifically suitable for publication and will be formally accepted for publication once it meets all outstanding technical requirements.

Kind regards,

Petri Kursula

Academic Editor

PLOS ONE
---

## [Editor Report · Acceptance letter]

8 Apr 2022

PONE-D-22-01714R1 

Crystal structure of an RNA/DNA strand exchange junction 

Dear Dr. Doudna:

I'm pleased to inform you that your manuscript has been deemed suitable for publication in PLOS ONE. Congratulations! Your manuscript is now with our production department. 

Kind regards, 

on behalf of

Prof. Petri Kursula 

Academic Editor

PLOS ONE